# Mixed sp^2^–sp^3^ Nanocarbon Materials: A Status Quo Review

**DOI:** 10.3390/nano11102469

**Published:** 2021-09-22

**Authors:** Jana Vejpravová

**Affiliations:** Department of Condensed Matter Physics, Faculty of Mathematics and Physics, Charles University, Ke Karlovu 5, 121 16 Prague, Czech Republic; jana@mag.mff.cuni.cz

**Keywords:** graphene, diamond, nanodiamond, diamane, graphene-diamond nanomaterials, all carbon nanomaterials, electrochemistry, mechanochemistry, sensor, supercapacitor, field-effect transistor, detector, superlubrication, tribology, graphene-diamond phase transformation

## Abstract

Carbon nanomaterials with a different character of the chemical bond—graphene (sp^2^) and nanodiamond (sp^3^)—are the building bricks for a new class of all-carbon hybrid nanomaterials, where the two different carbon networks with sp^3^ and sp^2^ hybridization coexist, interacting and even transforming into one another. The extraordinary physiochemical properties defined by the unique electronic band structure of the two border nanoallotropes ensure the immense application potential and versatility of these all-carbon nanomaterials. The review summarizes the status quo of sp^2^ – sp^3^ nanomaterials, including graphene/graphene-oxide—nanodiamond composites and hybrids, graphene/graphene-oxide—diamond heterojunctions, and other sp^2^–sp^3^ nanocarbon hybrids for sensing, electronic, and other emergent applications. Novel sp^2^–sp^3^ transitional nanocarbon phases and architectures are also discussed. Furthermore, the two-way sp^2^ (graphene) to sp^3^ (diamond surface and nanodiamond) transformations at the nanoscale, essential for innovative fabrication, and stability and chemical reactivity assessment are discussed based on extensive theoretical, computational and experimental studies.

## 1. Graphene and Nanodiamond—The Opposite Limits of Carbon at the Nanoscale

Carbon is the essential element on our planet; not only do all living species contain carbon, but also its pure forms exhibit unique properties. The arrangement carbon atoms can adopt in space can be very different, thus making possible a number of carbon allotrope structural variants (shown in Figure 1a) with differing properties arising from their particular chemical bonding and crystal structures. The hybridization of the carbon atom determines the most prominent types of arrangement—either sp^2^ giving rise to a strictly planar configuration or sp^3^ suggesting a tetrahedral coordination. The two extreme structures represent the most known allotropes: graphite and diamond.

The structure of diamond is built from carbon atoms which are covalently bound to the four nearest carbon atoms (an average bond length is 1.544 Å and the bond angle of 109.5° reflects the tetrahedral hybridization). Thanks to the covalent nature of the chemical bond and the ideal tetrahedral coordination, the diamond lattice shows extraordinary hardness, density (3.514 g/cm^3^), incompressibility and rigidity (designated as ten on the Mohs scale). It is one of the best conductors of heat, with heat conductivity up to five times higher than copper. Interestingly, diamond conducts sound, but it is an archetype insulator with outstanding optical transmissivity resistant to vast majority of chemicals. 

The lattice of graphite is already highly anisotropic at the level of the chemical bonding and the crystal structure. In the plane, the carbon atoms are connected in a honeycomb-like network of regular hexagons (average bond length and the in-plane bond angle are 1.418 Å 120°, respectively). These atomically thin all-carbon crystals are coupled together by weak van der Waals (vdW) interactions in the horizontal direction with an inter-layer distance of 3.347 Å. Due to the weak bonding between the layers, graphite is prone to cleavage. In contrast to diamond, it is less dense (2.266 g^−1^ cm^3^) and one of the softest inorganic solids (hardness is less than one on the Mohs scale). The crystal structure of graphite implies restriction of the electronic states in the basal plane, thus the charge carriers (electrons) conduct electrical current (and heat) only in the planes of the graphite structure and heat. In contrast to diamond, graphite strongly absorbs photons in the visible range of the electromagnetic spectrum.

In addition to the most prominent allotropes incorporating the two extreme hybridization possibilities, carbon atoms form additional configurations in space, especially when approaching nanoscale, see Figure 1a. Note that entirely new features may arise with a further downscaling of the nanocarbons, giving rise to the so-called carbon nanodots or quantum dots [1,2,3,4]. Thanks to quantum confinement, carbon nanodots are promising size-tunable photoluminescent materials

Nevertheless, the most explored nanocarbon species are graphene (GN) and nanodiamond (ND), directly derived from the structures of the macroscopic parent phases. The latter two types of nanocarbons are in the spotlight of current science thanks to their colossal application potential given by the unique fundamental physics behind them corroborated with multifaceted modification possibilities utilizing green chemistry.

Graphene is still believed to be a miracle material due to its straightforward structure [5]. Because of its atomic thickness, it is the thinnest stable crystal form on Earth and most likely in the universe. It is the lightest material, the strongest compound discovered (as the vdW bonds present in graphite are not involved), and the best heat and electricity conductor at room temperature. Thanks to its enormous thinness, the graphene shows unique levels of light absorption at about 2.3% of white light only and it shows outstanding flexibility. Graphene is the archetypic Dirac semimetal with a zero band gap (thus sometimes termed as zero-band gap semiconductor) and a linear dispersion of the energy bands at around the Dirac point [6,7]. Thanks to the unique band structure, the charge carriers (Dirac fermions) propagate through the material without disturbance, giving rise to carrier mobility up to 200,000 cm^2^ V^−1^ s^−1^.

Nevertheless, the term “graphene” is often used indistinctly for a plethora of two-dimensional (2D) sheet-like or flake-like nanocarbons, such as graphene oxide (GO), reduced GO (RGO), chemically modified graphene, few-layer graphene, etc. [8] Thus, a clear nomenclature for the two-dimensional (2D) nanocarbon materials should be applied; a systematic approach for the whole 2D nanocarbon family was defined by Bianco et al. [9]. 

Note that the various “graphenes” show entirely different physiochemical properties compared to “true graphene.” For example, GOs display bandgap variations from 2.00 eV up to 0.02 eV, depending on the degree of oxidation [10,11,12]. Consequently, the RGO shows a tunable absorption in the mid-infrared range by controlling its bandgap via the reduction level [11]. Also, the carrier mobility of the GO is much lower (0.05–320 cm^2^ V^−1^ s^−1^) in comparison with pristine graphene [13]. Nevertheless, all the graphene-based materials—despite lacking the outstanding properties of the true graphene monolayer—possess promising attributes, which make them excellent candidates for various applications.

For example, GO and RGO have entered the energy storage scene for the construction of batteries and supercapacitors [14]. The so-called laser-scribed graphene supercapacitors may offer power density comparable to high-power lithium-ion batteries, utilized today. Regarding other opportunities, sp^2^ nanocarbon-based materials have been employed in various types of sensors [15] and nanoelectronic devices [16,17]. A very efficient tuning of the 2D nanocarbons’ properties can be achieved by nitrogen substitution giving rise to a new family of carbon nitride networks [3,18,19,20,21]. The resulting C_3_N_x_ (X = 1–5) and C_2_N phases show promising photovoltaic, photocatalytic and even magnetic properties.

The properties of graphene and other graphene-related materials, however unique, are substantially influenced by their immediate environment—molecules in the surrounding atmosphere as well as atoms in the substrate transfer charge to the material. The extreme susceptibility to the ambiance can be a bottleneck and an excellent opportunity to boost these unique materials’ applicability further. Also, the quality of the nanocarbon-based material still prevails to be the limiting factor in all technological applications. 

Despite a tiny difference between the equilibrium structural arrangement of the graphite and the diamond by means of energetical demands, the latter is not favored at ambient conditions, regardless of downscaling the particle size to nanometer dimensions [22]. However, the outstanding physio-chemical properties of the diamond make this carbon allotrope also a highly demanding material for various applications.

NDs represent a unique case among the nanomaterial as their properties comprise outstanding mechanical performance, sufficient chemical inertness, excellent biocompatibility, and exceptional optical and optoelectronic properties, significant at the visible part of the electromagnetic spectrum [23,24]. The bandgap in NDs varies with the NDs’ size and surface functionalization [25] through external (C–H substitution) or internal (replacing CH or CH_2_) doping. Doping with electron-donating or withdrawing groups (sometimes termed as “push-pull”) reduces the bandgap to that of bulk diamond (~5.5 eV). Further reductions down to 1–2 eV can be achieved with charged substituents. Increasing the size of the NDs and “push-pull” doping is the best strategy to tune the NDs’ properties for various applications, mostly in nanoelectronics [25]. Comparing to graphene, the sp^3^ carbons show in general much lower charge mobilities. Single-crystal diamond exhibits both the highest electron and hole mobilities at room temperature of any wide-bandgap semiconductor of about 4500 cm^2^ V^−1^ s^−1^ and 3800 cm^2^ V^−1^ s^−1^, respectively [26,27]. In the ND films, the mobility depends on the amount of the sp^2^ carbon phases occurring at the grain boundaries and doping (N, B) [28]. Highly conductive N-doped ND films possess high electron concentrations, of up to 10^20^–10^21^ cm^−3^. In contrast, the electron mobility is between 1 and 10 cm^2^ V^−1^ s^−1^, which is somewhat comparable to amorphous silicon.

NDs also provide unique spectral features unattainable in the molecular world, such as color centers embedded in NDs’ crystal lattices, suggesting their use in diagnostic and imaging approaches in biomedicine as sophisticated optical tools. This nitrogen-vacancy (NV) centers can respond to changes in the nanoparticle environment with exceptional sensitivity and report on various local variables. Thus the NV centers can be used to construct attractive nanosensors because NV center offers a unique sensitivity to the extremely weak electric and magnetic field at a nanoscale resolution [29,30,31].

Like in the case of the graphene, the performance of NDs is strongly influenced by structural features such as the particle size, shape, crystallographic order, chemical functionalization of the surface layer, level of intrinsic defects (vacancies, stacking faults, impurity atoms, etc.), and presence of graphenoic carbons in the particle shell [24].

In this vein, mitigation of the effects of physio-chemical limits and sample quality by creating novel all-carbon nanomaterials appeared to be a very promising direction. Owing to the exceptional properties of sp^2^ GN (like Young’ modulus, strength, thermal and electrical conductivity, etc.) and sp^3^ diamond and ND (chemical stability, hardness, thermal conductivity), it is also appealing to combine those nanocarbon allotropes in an advanced cohort of future nanoelectronics and optoelectronic devices, various functional composites, heterostructures, and networks. The most promising directions are summarized in the scheme presented in Figure 1b.

This review aims to summarize important aspects of the current graphene—diamond nanomaterials’ research. The structure of the review is as follows. First, phase transformation between the graphene and diamond at the nanoscale is discussed in Section 2. This section also concerns mutual phase transformations, usually when a diamond crystal or a ND thin film is used for a homo-elemental growth of GN and vice versa. Second, graphene–diamond heterostructures developed for sensing applications are described in Section 3. Third, the GN-diamond heterojunctions are discussed from the point of view of tribology and nanoelectronics in Section 4. GN—diamond-based nanomaterials developed for other exciting applications such as energy storage, materials’ processing, detectors, and light sources, catalysts, and nanoscale pressure devices are summarized in. Finally, a summary and future prospect of the GN–diamond nanomaterials are given in Section 6.

## 2. Graphene—Diamond Phase Transformations at Nanoscale

Although the energy difference between the parent bulk carbon allotropes is very tiny (about 0.02 eV/carbon atom only), an activation barrier required for the phase transformation is 0.4 eV, and the conversion of graphite to diamond occurs under extreme pressures and temperatures. However, for ND particles (3–6 nm), tetrahedral coordination is preferred, making the stabilization of NDs easier [22]. Thus, the sp^2^–sp^3^ transformations at the nanoscale are the most important processes governing the final materials’ stability, reactivity, and physical properties. Both the theoretical and experimental investigations pointed to the three crucial aspects of the transformation: the initial configuration of the sp^2^–sp^3^ carbon fragments, number and stacking of the GN layers, and chemical functionalization of the GN and diamond surfaces. Selection of first principle studies and modeling, and experimental conditions, and difficulties of the peculiar sp^2^–sp^3^ all-carbon phase transitions will be discussed in this section.

### 2.1. First Principle Calculations and Modeling

There are two possible types of the sp^2^ to sp^3^ transformation at the nanoscale; GN to diamond fragment (or ND-like structure) and vice versa. First, the theoretical findings on the conversion of GN and derived systems to diamond-like structures will be discussed. 

Kvashnin et al. explored the phase transformation of a hydrogenated few-layer GN into a thin ND layer at low-pressure conditions [32]. The authors concluded that due to the size confinement at the surface, such a “chemically-induced phase transition” is strongly dependent on the thickness of the initial GN source. They also suggested a pressure-temperature phase diagram of quasi-2D diamond—multilayered GN system, which accurately predicts the relationship between the film thickness and thermodynamic parameters, which are essential for evaluating the feasibility of creating novel quasi-2D materials. 

Paul and Momeni also studied the mechanochemistry of the hydrogenated few-layer GN and diamond thin films as a function of the layer thickness, pressure, and temperature [14]. In agreement with other studies, they also observed that the phase transformation is driven by the chemical functionalization and number of GN layers. Although the pristine multilayer GN to diamond conversion seems to be reversible, hydrogenated multilayer GN tends to transform into a diamond-like layer, which is metastable. 

Antipina and co-workers reported various chemical functionalizations of few-layer GN with H (H_2_), F (F_2_), NH_3,_ and H_2_O induces spontaneous conversion to diamond films with a specific crystal structure (cubic or hexagonal). The study suggests that by careful control of the type and amount of the functional groups in the initial GN source, diamond layers with well-defined surface properties can be reached [33]. 

Belenkov et al. suggested that crosslinking of GN sheets may lead to the stabilization of various diamond-like nanostructures [34]. The authors also predicted X-ray powder diffraction patterns, which are important for the unambiguous identification of these new structural arrangements in experiments.

Reactive molecular dynamics simulations revealed that bilayer GN could also undergo specific structural re-arrangements into a diamond-like structure [35]. These conversions are strongly dependent on the number of layers in the original few-layer GN and also on their mutual stacking. For example, if the initial system is composed of three- to six layers in AB-type stacking, the transformation does not occur up to extreme pressures of 200 GPa. The study also suggests that the ABA type of stacking tends to suppress the transformation into the diamond phase. On the contrary, for precursors with ABC and ABCAB types of stacking, the transformation is favored. Thus, the “stacking faults” present in the few-layer GN with more complex stacking patterns promote the high-pressure conversion of multilayer GN to diamond [35].

Also, it has been demonstrated that the diamond (literally speaking a part of the diamond surface) can be transformed into GN and related nanostructures. 

Alekseev investigated the case when a copper film (known as a catalyst for the GN growth) approaches its melting point, and a crystalline surface of a diamond is in contact with the molten metal. His quantum-chemical simulations revealed that the upper layer of carbon atoms of the diamond surface tends to reorganize into a GN lattice [36]. He reported that the carbon atoms just below the affected top layer of the diamond surface move to the equilibrium positions due to the thermal activation, corresponding to the stable GN arrangement.

Also, the formation of another GN arrangement, induced via surface reconstruction—GN-like stripes on a diamond C(331) was predicted [37]. Van Wijk and Fasolino demonstrated that the transformation of diamond to amorphous carbon could be blocked by adding GN spacers between the sp^3^ slabs; two GN layers are already sufficient [38].

The GN—diamond transformations may also yield exotic novel phases, see Table 1 and Figure 2. For example, Cellini et al. found that two-layer GN behaves as efficient protection, showing exceptional mechanical performance under pressure. The authors carried out detailed DFT calculations, which revealed that a reversible sp^2^ to sp^3^ transformation occurs under compression, which is the reason for the enormous mechanical resistance. Also, their calculations yielded evidence of the formation of a unique diamond-like monolayer structure termed diamane [39] (Figure 2a). 

Erohin et al. carried out atomistic first-principles calculations to predict the thermodynamic and kinetic parameters essential to stabilize the diamane phase [40]. They also reported that the thickness of the GN precursor is a very important parameter for the conversion. For example, at the bilayer limit, the system tends to transform into the diamane, while for more layers, the stable product is typically lonsdaleite. The important structural parameters are the conformation of the adsorbent pattern, which is the chair vs. boat for the bilayer and few-layer GN, respectively. The authors also addressed the role of chemical functionalization. The suggested that the small atoms, like H and F, catalyze the reconstruction into diamond, while larger atoms, such as Cl, block the transformation due to sterical demands.

**Table 1 nanomaterials-11-02469-t001:** Exotic nanocarbon phases and architectures due to graphene-diamond reconstruction at the nanoscale.

System	Evidence	References
Diamane	Atomistic first principles computations	[41]
F-diamane	HRTEM, EELS	[42]
H-diamane	Optical absorption, XRD	[43]
LA10 phase	DFT	[44]
Graphene Arch-Bridge	First principle calculations	[45]
Diaphite	DFT, HRTEM	[46,47,48]

Several authors have studied the effect of introducing a non-carbon atom on the sp^3^–sp^2^ transformation. The role of B doping on GN formation on diamond (111) surface was studied by Gu et al. [49]. For a specific level and spatial distribution of the B atoms, the GN tends to grow spontaneously. A similar problem was addressed by Lu et al. [50]. The diamond surface reconstruction due to introducing B atoms was also identified as the underlying mechanism for the GN stabilization on the diamond surface.

Another case was explored by Okada, who investigated the formation of GN on diamond nanowires by first-principle total-energy calculations [51]. He suggested that multi-layer GN develops due to graphitization on the surface of the wire. The resulting all-carbon core-shell wires are indirect semiconducting with a small band gap.

A very interesting computational study by Greshnyakov et al. suggests the formation of a diamond-like phase (termed LA10, I4_1_/amd space group), in which all the carbon atoms are accommodated in symmetrically equivalent crystallographic positions [44]. The exotic LA10 phase can be obtained by pressurizing graphite. The special arrangement of carbon atoms in this structure can be viewed as an infinite multiplication of L4–8-type GN layers with tetragonal symmetry, consisting of four-membered and eight-membered units. Band structure calculations revealed that the exotic LA10 phase is semiconducting with a wide band gap (5.0 eV to 6.1 eV).

Another strategy to reach the GN-diamond transformation counts on expansion and cooling the carbon phases. Ileri et al. suggested that NDs can be stabilized when carbon precursors expand and cool to normal conditions [52]. Their quantum simulations investigated such processes during shock compression and revealed that depending on the conditions, the plethora of ring and chain fragments could be obtained. Their type and ratio determine the relative content of GN and ND in the cooled product. In general, the GN flakes dominate [52].

**Figure 2 nanomaterials-11-02469-f002:**
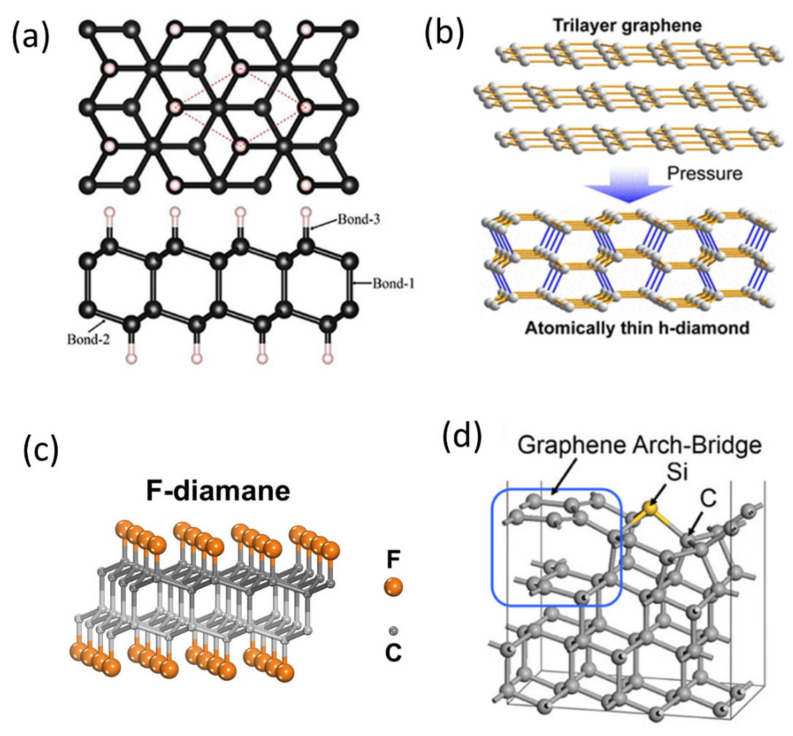
Examples of graphene—diamond transformations at nanoscale. (**a**) fully optimized structure of diamane. Reprinted with permission from [53]. Copyright 2020 IOP Publishing. (**b**) Pressure-induced graphene to H-diamane transformation. Reprinted with permission from [43]. Copyright 2020 American Chemical Society. (**c**) F-diamane (adapted from https://www.techexplorist.com/converting-graphene-diamond-film/28444/, accessed on 30 June 2021), and (**d**) Graphene arch-bridge. Reprinted with permission from [45]. Copyright 2020 American Chemical Society.

The mechanism and key parameters of the diamond-like phases deposition on top of multilayer GN was investigated by molecular dynamics simulations [54]. It was demonstrated that the incident energy decrease reduces the adhesion of the NDs on the GN surface. This is evidenced by the increase in the number of dangling bonds at the interface GN-ND interface. The authors also concluded that the diamond-like phases and nanostructures grown on the GN-type surface have a reduced content of the sp^3^-type carbon atoms comparing to those grown on a surface of a diamond, regardless of the crystallographic orientation. Another important criterion is related to the fragments, which may form ordered or disordered ring-like assemblies. Diamond-like structures formed on the sp^2^-type surface typically consist of disordered ring-like constituents.

Recently, a new type of hybrid sp^2^–sp^3^ carbon nanomaterial, termed diaphite, was reported [46]. In fact, the diaphite represents a class of nanocomposite-like all-carbon compounds of both natural and artificial origin, which have structural patterns very similar to the lonsdaleite and other carbon phases formed via shock processing of the graphite. Thanks to the unique coexistence of the sp^2^–sp^3^ carbons forming a kind of ND-sp^2^ network composite, these materials exhibit exceptional hardness but simultaneously a significant ductility; this is why they are sometimes called “ductile diamonds.”

In addition to the long-range and nanocomposite-like structures, nanosized architectures linking the structure of graphene and diamond were also reported. Bai and co-workers suggested the formation of the so-called “graphene arch-bridge” (Figure 2d). This interesting architecture of carbon atoms can be stabilized on a diamond (111) surface by Si doping. Introducing the Si creates substantial stress at the position of the substitution in the lattice; the sp^3^ to sp^2^ transformation occurs locally. Thus, the initially stable six-member rings of carbon atoms transform into the five-membered rings. The authors also proposed that the “graphene arch-bridge” is responsible for the unusual tribological properties of Si-doped NDs and diamond-like nanomaterials. 

### 2.2. Experimental Demonstration and Growth Mechanism

Up to now, the large number of first-principle and computational studies presented in the previous subsection prevails over experimental studies on this phenomenon. Also, formation of an atomically thin diamond crystal is not possible due to the sp^3^ hybridization, which would make such a crystal highly unstable. Thus, anchoring other chemical species on the diamond surface is necessary to stabilize such a 2D architecture. 

A first convincing experimental proof of such a conversion was delivered by Bakharev and co-workers [42]. They observed that the fluorinated AB-stacked bilayer GN grown on a CuNi(111) surface by the CVD yields a fluorinated diamond monolayer (‘F-diamane’), see Figure 2c. A similar structure, the so-called H-diamane, was obtained by pressure-induced transformation of few-layer GN [43], see Figure 2b.

Another strategy to induce formation of GN on diamond is the use of a high-temperature treatment and a catalyst and high-temperature treatment; typically, copper or nickel, which are the most common for the CVD growth of graphene. Carvalho reported on simultaneous growth of GN and NDs by the microwave plasma CVD with the help of copper catalyst (standard for the CVD growth of GN). The authors reported a detailed HRTEM study, which enabled the formulation of a diamond-on-graphene nucleation mechanism [55]. The copper catalyst was also used in the high-temperature annealing of a diamond, which yielded GN layers of very good quality [56].

Ordered GN films have been grown on Fe-treated diamond and silicon carbide (SiC) surfaces; this method enabled the epitaxial growth of GN on the diamond for the first time. Also, the GN films formed only on the catalyst-treated regions, enabling patterning the GN on diamond (and SiC) at temperatures reachable in industrial technologies [57,58]. 

Multilayer GN can also be grown using another catalyst, widely used in the CVD growth—nickel. For example, a thin nickel film was deposited by e-beam evaporation on diamond (001) substrates, and the multilayer GN precipitated upon cooling. As in the case of the CVD of GN on a nickel foil, the growth occurs due to the dissolution of carbon in the nickel and subsequent segregation of the large-area GN films on top of the nickel catalyst [59]. Using a similar strategy, GN films were also obtained on diamond (111) surfaces via annealing [60]. Also, Berman and co-workers achieved the transformation of polycrystalline diamond into GN using a rapid thermal [61]. In addition, they demonstrated the growth of free-standing GN, which is a significant technological step forward for large-scale GN/diamond-based electronics production. 

Surface transformations of carbon deposited on polycrystalline nickel by hot filament-CVD were experimentally investigated by Rey and Normand [62]. The process starts with a formation of graphitic layers and GN, then graphitic/carbidic and diamond-like carbon phases are stabilized. Finally, a rapid transition to the diamond occurs. The transformation is strongly affected by the character of the nickel surface, which can be efficiently controlled via annealing.

Taking advantage of the Ni-assisted diamond transformation, Tsubota et al. proposed a new simple method for fabricating a microchannel covered with a GN layer on a diamond substrate [63]. Also, Ni/Fe catalyst-assisted process in a vacuum induction furnace enables growth of GN nanowalls directly on diamond particles [64]. 

Interestingly, the GN can also promote the growth of another carbon-containing phase on the diamond. Pure Fe catalyst was used to persuade the growth of SiC nanowires on a diamond surface upon heating at 1300 °C in the presence of GN as the second carbon source. The authors also suggested a model of the GN-assisted growth of the SiC phase on the diamond crystalline surface [65]. 

Also, several works report on the growth of diamonds on GN and GN-like surfaces. Li et al. observed that highly crystalline diamond can be grown on GN via a quasi-homoepitaxial growth, which requires the sp^2^–sp^3^ hybridization and covalent bond transformation [66]. The process has several stages, including the formation of a carbon network coupled via non-covalent interactions, which promotes the conversion of GN into the diamond. Because the precursor network is a microporous assembly of carbon atoms, highly porous diamond layers with the hydrophilic surface can be obtained by this method. Thus, the strategy seems to be rather promising for the growth of various GN-diamond nanomaterials with specific surface properties and porosity.

Hussain et al. suggested that applying the Hummer method on GO combined with an ultrasound exposure (several minutes to few hours) can result in GN oxide (GO) formation and ND formation, respectively [67]. The diamond-like carbon can also be transformed into GN and reduced GO by nanosecond pulsed laser annealing [68].

Hembram et al. achieved a carbon-pure formation of the diamond via surface hybridization with GN [69]. GN multilayers (5–50 nm in thickness) were grown vertically onto a polished <110> textured polycrystalline diamond film via hydrogen plasma etching in a CVD chamber at ~1300 °C. A fascinating analogy of the surface defects known for the diamond, the hybrid GN-diamond interface contains “interface defects” responsible for a strong photoluminescence signal. Also, this all-carbon surface hybrid reveals a finite bandgap, which scales with the number of GN layers involved in the structure.

Interestingly, a new type of diamond-based nanomaterial—the so-called diamond aerogel was obtained from GN aerogel by high-pressure laser treatment in a diamond anvil cell [70]. The morphology of the diamond aerogel was found to be predetermined by the microstructure of the graphene precursors. 

Another study dealing with the thermobaric treatment of mixtures of carbon-containing phases was reported by Filonenko et al. [71]. The authors concluded that the transformation of carbon nitride to a diamond-like structure occurs without a need for a catalyst at such extreme conditions. Similarly, Fukuda and co-workers also suggested that a high-temperature, high-pressure (HTHP) crystallization of 2D-GO reveals 3D diamond-like nanostructures [72]. 

The mechanisms behind the GN-diamond transformations were also investigated by photoelectron spectroscopy [73]. In this study, the (111) phase of the diamond was annealed in a vacuum. The authors determined the onset of the graphitization at ~ 1120 K. They concluded that in analogy with the SiC, a buffer layer composed of sp^2^-carbon atoms develops at the interface between graphene and the diamond surface. 

The formation of a graphene-on-diamond structure by the graphitization of a diamond (111) surface was also reported by Tokuda and co-workers [74]. The authors observed that the process consists of two steps—formation of an atomically flat diamond (111) surface by homoepitaxial lateral growth and graphitization.

Besides the structural aspects of the GN-diamond transformation, a cross-over between the semiconducting and metallic properties in GN—diamond-like carbon heterostructures were investigated by Zhao and co-workers [75]. The results show that different GN terminations (N, F) strongly affect the electronic properties of the GN nanostructures. The hybrids with F-functionalization revealed p-type doping with high mobility, while the N-functionalized nanostructures exhibit almost metallic carrier densities.

A rare attempt to grow diamond on tungsten was found to be enabled by GN [76]. In this process, the nucleation of the diamond is heterogeneous, and the formation of sp^3^-type C-W bonds is essential for the diamond phase stabilization. The role of GN is to serve as the source of carbon and nucleation center; the growth of the diamond film propagates from the edge of the GN on the tungsten. 

Another unusual alternative to producing GN on the diamond surface exploits low-energy Ar ion beams [77,78]. The Ar ions create defects in the diamond lattice, which leads to the formation of an amorphous carbon layer. The disordered carbon surface tends to stabilize by crystallization to the GN. From the point of view of industrial scale-up, the advantage of the process is the low temperature of the treatment. 

## 3. Graphene—Diamond Sensors and Biosensors

GN and GO based electrodes combined with NDs and more complex nanomaterials based on the all-carbon networks were identified as great and versatile platforms for the construction of various types of sensors. Among them, the most recognized are the electrochemical sensors and biosensors. An overview is given in Table 2, and selected cases are discussed hereafter.

### 3.1. Electrochemical Sensors

Electrochemical sensors based on GN-diamond hybrid nanomaterials are highly competitive for various applications. Electrochemical determination of nitroaromatic explosives using boron-doped ND(BDND)/GN electrodes was proposed by Dettlaff and co-workers [84,95]. The morphology of the electrodes features a unique nanowall-like structure of the GN component, and it shows superior electrochemical performance with the detection threshold of 0.52 ppm thanks to the complex sp^3^–sp^2^ composite surface. 

Another type of high-performance GN/diamond electrode was reported by Gao et al. [85]. The sensing platform features a highly robust design with a high degree of regeneration—a simple sonication can achieve this. The system was designed to resolve enantiomers in the beta-cyclodextrin drop-casting process, successfully resolving the D- and L-phenylalanine. Therefore, the proposed GN/diamond is a promising system for various enantioselective sensing applications.

Also, a series of ND and GN hybridized films with various morphologies and compositions of grain boundaries was investigated for application as an electrode material [96]. The hybrid films were grown by hot-filament CVD and their phase composition and morphology varied with the pressure maintained during the deposition. The main benefit of these films for electrochemical applications is the balance between the diamond (enlarging the potential windows and decreasing the background current) and the GN (improving the redox current) components. The morphological nature of the GN helps to improve the electrochemical response as the sp^2^-component is very well ordered and forms a thin layer on the diamond grains.

The same collective of authors developed another powerful hybrid electrode system with three interfaces (air-solid-solution) [83]. The system revealed an excellent performance in the carbon-based oxygen-reduced reaction (ORR). The morphology of the electrode is very complex; it consists of vertically ordered GN flakes, which are adhered to well-separated ND grains. Interestingly, the edges of the GN component contain a high number of structural defects, which serve as highly efficient catalytic centers. The authors also boosted the original design for higher performance by careful control of the hydrophobicity. They demonstrated that hydrophobic to the super-hydrophobic surface can be efficiently tuned by the tilt of the vertical graphene flakes. The study thus suggests a simple and smart strategy to tailor catalytically active GN/ND films.

Hybrid systems based on GN and diamond containing another functional nanomaterial have also been recognized. For example, a plasmonic hybrid composed of a 2D assembly of gold nanostructure on a diamond-like film was prepared by two-step electrodeposition from RGO [82]. During the process, the redox properties of the RGO surface were controlled by changing the electrochemical reduction time. Consequently, the number of nucleation sites for the growth of gold nanoparticles was optimized. The plasmonic hybrid was tested for surface-enhanced Raman scattering (SERS) applications using the standard molecule, rhodamine B. The obtained enhancement was about 860 (normalized to the Si reference signal). The very high enhancement factor suggests a synergy of the SERS’ electromagnetic and chemical enhancement mechanisms [97]. The platforms have great promises for utilization in various biochemical, environmental, medical, and food safety applications.

Another example of a GN-diamond hybrid with plasmonic metal is a new electrochemical sensor designed by modifying the commercial boron-doped diamond electrode with GO reduced electrochemically and further electrode coated with silver (Ag/GO/BDD) electrode was selected to detect tetracycline in an aqueous solution.

Pei and co-workers reported a sensing platform for the electrochemical detection of trace Pb^2+^ in salt-water composed of the boron-doped diamond electrode and GN [88].

Besides the plasmonic components, organic compounds and polymers have also been incorporated in the GN-diamond nanomaterials for electrochemical sensing. A nanocomposite with polyaniline was prepared for the electrochemical determination of 2,4dichlorophenol in aqueous media [89]. The preparation of the active electrode started with oxidative polymerization of aniline in the presence of GN and diamond. The resulting nanocomposite was then deposited on a glassy carbon electrode (GC). The electrochemical performance of the bare GC electrode was compared to that modified by the nanocomposite; the latter revealed superior detection abilities.

Peng et al. [90] developed a complex nanomaterial based on nitrogen-substituted ultrananocrystalline diamond (UNCD) and multilayer GN. The nanocomposite films were grown by plasma-assisted CVD from diethylamine. The structure of the film is rather complex; the GN flakes are aligned vertically, and the high-order structure is highly porous with a nest-like morphology. Thanks to the unique architecture, these films are very robust against cycling and show a substantial electrochemical double-layer capacitance (EDLC). Their unique properties are beneficial not only in electrochemical sensing but also as supercapacitor materials, etc.

Alternative electrode materials for detection of carbaryl and paraquat pesticides selectively from aqueous solution were also investigated [91,92]. For that purpose, GN-modified boron-doped diamond electrode was compared to the same type of electrode, boosted electrode with Ag. The incorporation of Ag significantly improved the electrocatalytic activity towards carbaryl electrooxidation and paraquat electroreduction [91]. Finally, the practical application of these electrodes was verified on a series of samples containing water from different resources.

To complete the puzzle of the outstanding electrochemical properties of the diamond-GN nanomaterials, the RGO/boron-doped diamond electrode with excellent electrochemical oxidation performance also revealed antibacterial activity [98]. 

### 3.2. Biosensors

Hybrid GN-diamond-based nanomaterials are also convenient for the fabrication of various types of biosensors. Cui et al. reported on the fabrication of precise and robust devices for glucose detection composed of nickel nanoparticles embedded in the GN-diamond network [81]. The sp^3^-sp^2^ electrode material was fabricated using the catalyst-assisted strategy described in Section 2.2. The authors used nickel films to promote the growth of GN on the diamond surface. During the growth, nickel particles nucleate in situ due to the retraction of the melt from the non-wettable GN surface. The presence of the nickel nanoparticles improves the charge transfer to the Gn component during the electrochemical reaction. This biosensor was found to be highly selective; thus, it has a solid potential for accurate monitoring of glucose in blood without compromising the detection limits. 

Electrochemical sensing of L-glutamate with the help of a CVD-grown diamond electrode with incorporated GN nanoplatelets was reported by Hu and co-workers [80]. In addition to the carbon constituents, the electrode also contains platinum nanoparticles, which act as a catalyst of the electrooxidation processes employed in the detection. The selectivity of the biosensor was improved by adding a layer of polyphenylenediamine.

Huang and co-workers reported that RGO and boron-doped diamond show an excellent response to the various protein adsorption, which is mirrored in the impedance variation [79]. The authors also reported that the adsorption and the orientation of the protein molecules on the surface could be controlled using the RGO/boron-doped diamond electrode.

GN-modified boron-doped diamond electrodes were explored for electrochemical detection of epinephrine in the presence of uric acid [86]. The best sensitivity toward epinephrine oxidation was obtained at pH 7. The authors also reported that the GN helps to improve electrochemical oxidation. 

Boron-doped GN and diamond electrodes were also tested for oxidative sensing various biomarkers [93]. Significant differences in the oxidation of ascorbic acid, uric acid, dopamine, and β-nicotinamide adenine dinucleotide were reported. On the other hand, both materials exhibit comparable sensitivity towards ferro/ferricyanide.

Yuan et al. fabricated an electrode for electrochemical detection of dopamine. In the fabrication process of the electrode material, they converted the surface fraction of a diamond into a few-layer GN by HPHT catalyst-assisted procedure [94]. Also, they reported that the fouled electrode surface could easily be cleaned by ultrasonication, which makes this type of mixed carbon electrode very competitive for real applications in biosensing.

## 4. Graphene—Diamond Interfaces and Heterojunctions

The interface between the sp^2^ and sp^3^ nanocarbons, where the two nanoallotropes are in intimate contact, is the most exciting entity in the GN-diamond-based nanomaterials. It ensures the two-way communication channel for the charge, spin, molecule, and phonon transport and provides the fragile stability of the carbon atoms’ in the desired configuration. 

The rather weak interaction between the two nanocarbon allotropes determines the unique mechanical performance of these hybrid materials thanks to sliding-induced graphitization and passivation of dangling bonds [99]. Thus, the interface is responsible for the GN-diamond heterostructures’ outstanding lubrication and mechanical properties; this topic will be discussed in the first part of this section. 

From the view of band structure renormalization, the all-carbon GN—diamond interaction has been widely addressed with DFT calculations (e.g., [100,101]). The results show consistently that vdW interactions dominate in the GN—diamond interaction. Consequently, the carrier mobility of GN is almost intact, except for a tiny bandgap opening at the Dirac point. Nevertheless, the charge transfer between the GN and the diamond depends on the phase orientation of the diamond surface. While no charge transfer between GN and diamond (100) surfaces occurs, p-type or n-type doping on GN can be observed on the (111) surface. The interaction can be further modified via diamond doping; nevertheless, the GN keeps its aromatic character [101]. Thus, GN adsorbed on the (111) diamond surface shows a finite gap depending on the adsorption arrangements due to the variation of on-site energy, keeping the linear band dispersion [102]. Also, the electronic spin arises due to the exchange proximity interaction between GN and localized states in the diamond. These predications make the GN–diamond heterojunction as a viable platform for future GN-based nanoelectronics applications, which will be discussed in the second part. 

### 4.1. Friction, Tribology, and Mechanical Properties

Superlubricity is a unique tribological property with enormous application impact. In this special regime, the friction between the phases in interactions vanishes, which literally means that the kinetic friction coefficient decreases below 0.01. Carbon-based materials show promising results towards achieving extreme lubricity for various industrial and technological applications. The main reason for their excellent performance is the on-surface reconstruction of the carbon network by means of a tribochemical reaction.

For example, oil lubrication induces structural changes in hydrogen-free tetrahedral amorphous carbon [103]. This carbon nanomaterial contains sp^3^-hybridized carbon primarily, but the lubrication causes the formation of extremely thin (~1nm) partially oxidized GN sheets, responsible for the close to superlubricity regime. The appearance of these mesoscopic GN-derived structures was also confirmed by atomistic simulations, which revealed the general relation between the microscopic tribochemical mechanism and the macroscopic lubricity in the nanocarbon—oil systems. 

It has been also demonstrated that using diamond brings the experimental realization of friction-free GN one step closer [104]. Berman et al. reported reducing macroscopic friction for a system composed of GN flakes and NDs [47]. The hybrid all-carbon nanomaterial was deposited on a silica surface, and the friction performance was studied. The researchers observed that the extreme decrease of the friction coefficient is due to forming GN-ND nanoscrolls as the GN flakes are sliding on the silica surface and tend to catch and wrap the ND particles (see Figure 3). The extreme value of a friction coefficient (~0.002) was recently reported for Si-doped hydrogenated amorphous carbon lubricated with GO-ethylene glycol dispersion [105]. 

The lubricity of hydrogenated diamond-like films was investigated by Liu et al. [106]. In agreement with the work of Berman and co-workers [47], the authors concluded that low friction occurs due to the nanoscrolls formation. The authors also reported that to minimize the friction coefficient, the graphitization of the surface can be controlled via the sliding mode. 

The graphitization due to the tribological transformation at the interface in GN-diamond films was explored by Chen et al. [107]. The authors reported that the diamond film’s surface morphology is essential for reducing the friction coefficient due to the formation of the nanoscrolls, also observed by the groups of Liu or Berman. The authors also reported this GN/diamond coating friction and wear behaviors via sliding tests [108]. 

The relevance of GN-coated diamond thin films as lubricants was studied by Shen and co-workers [109]. The same collective also elucidated the atomistic mechanism behind the experimentally observed low friction of these all-carbon lubricants [110]. The low friction at the GN-diamond interface was also proposed by DFT calculations [99] and by molecular dynamics simulations [111]. The theoretical studies suggested that the best lubricity is attained when the GN slides along its armchair direction.

Also, hydrogenated multilayer GN on diamond-like films revealed excellent lubrication properties, as suggested by molecular dynamics simulations [112]. Moreover, the improved lubrication performance is driven by the adhesion to the substrate. Another important aspect is the rigidity of the hydrogenated GN layer, which influences the friction due to varying roughness at the level of individual atoms. This effect clearly correlates with the level of hydrogenation. Thus, the study introduces a strategy for the rational design of outstanding lubricants based on chemically functionalized GN. 

Also, the adhesion of a GN monolayer onto a diamond (111) surface with born and nitrogen doping was investigated by first-principle calculations [101,113]. The authors concluded that the GN kept its aromatic character for all terminations and only a moderate charge transfer from the GN to the diamond surface occurs. 

A comparative study of the tribological performance of diamond-like/ionic liquid films upon different sp^2^ carbon additives for aerospace and space engineering was reported by Zhang et al. [114]. The authors observed that various lubrication effects could be achieved due to very different tribological mechanisms determined by the additive (GN, MWCNTs).

Mechanical properties of the GN-diamond interface were also addressed by Machado et al. by molecular dynamics simulations [115]. They proposed a system composed of ND superlattice embedded in twisted bilayer GN and observed that the mechanical properties of these superstructures could be tuned by controlling the fraction of sp^3^-hybridized carbon atoms. The authors also confirmed that the non-planarity of the carbon network could be sustained via a chemical functionalization. 

The stability, elastic moduli, and deformation behavior of GN-diamond-like nanomaterials were studied employing molecular dynamics [116]. The non-elastic deformation in these sp^2^–sp^3^ networks can be understood via a change of the bond length and angles.

An interesting problem related to the GN-diamond lubrication properties is the atomic force microscopy (AFM) experiment using a sharp diamond tip. Molecular dynamics simulation predicted the friction variations for different GN morphologies for the diamond tip’s different shape and adhesion parameters [117].

### 4.2. Nanoelectronic and Spintronic Platforms

Electronic properties of GN-diamond heterostructures attracted considerable attention due to their potential exploitation in nanoelectronics, optoelectronic, and spintronic devices. Besides the GN-single crystal diamond heterostructures [118,119], the properties of nanocarbon hybrids were also investigated. 

GN sheets decorated with ND particles have been investigated by Wang and co-workers [120]. At the ND-bonded regions, the carbon atoms attributed to the GN layer follow sp^3^-like bonding; these sp^2^–sp^3^ junctions represent conduction bottlenecks for the percolating sp^2^ GN network. The low-temperature transport measurements revealed an insulating behavior associated with Anderson-type localization for the charge carriers. Also, a significant negative magnetoresistance was observed, which can be attributed to the magnetic correlations of the localized charge carriers due to extrinsic metal (magnetic) impurities associated with the ND.

Hu and co-workers studied n-type UNCD/GN films with implanted oxygen ions yielding C=O termination of the ND grains giving rise to a conductive network surrounding the ND crystallites. The hybrid films show high n-type Hall, making them efficient construction materials for various nanoelectronics and field emission devices and electrochemical electrodes [121].

Bogdanowicz and co-workers reported on the first step towards a diamond-based transistor. For the device fabrication, they used a thin boron-doped diamond layer grown on a tantalum foil transferred to GN/Si/SiO_2_ [122]. The resulting device exhibits thermionic conductance and variable hopping above and below 50 K, respectively. 

Hu et al. investigated the electronic interaction between GN and semiconducting diamond substrate by DFT calculations [100]. The authors reported that the most significant feature of these interactions is the formation of charge-transfer complexes at the interfaces between the GN and (111) phase of diamond, inducing either p- or n-doping of the GN. Based on the specific surface orientation, the single-crystalline diamond is a convenient platform for GN-based nanoelectronics devices. The same collective of authors also applied first-principles calculation to explore the diffusive thermal conductivity of diamane [53].

Selli et al. also investigated the interaction of a GN monolayer with the diamond using DFT and quantum transport calculations [123]. They observed that the deviation of the GN from the planarity due to the GN-diamond bonding affects the transport properties. The hybrid GN-diamond system forms a net of periodic ridges, which introduce high anisotropy to electrical transport. The carrier mobility is much higher along the ridges comparing to the low mobility in the perpendicular direction.

More complex systems—fluorinated GO/ND thin films have been proposed for electromagnetic restraining applications [124]. In addition, these nanomaterials revealed promising optical properties for various optoelectronic applications. The hybrid nanocarbon materials for fast optoelectronic devices were also investigated by Konabe et al. [125]. The authors investigated multiple exciton-generation in GN nanoribbons using the tight-binding method. They show that the ribbons with armchair termination show photoelectric conversion ~100%. 

Diamond-like films, obtained by the cathodic vacuum arc deposition method, were used as a dielectric platform to support GN [126]. The GN deposited on these films revealed improved FET carrier mobility compared with GN on SiO_2_/Si substrate due to the significant suppression of the doping. GN transistors operating at high frequencies were successfully fabricated on the diamond-like substrates by Wu et al. [127]. 

The GN-diamond heterostructures were also explored in the direction of spintronics. Gap opening and spin injection was investigated by Ma and co-workers [102]. They addressed GN’s electronic and magnetic properties on the (111) diamond surface by the first-principles calculations and concluded that in such a configuration, a finite gap opens, and its magnitude changes with the actual carbon configuration on the diamond surface. At the same time, the linear dispersion of GN is preserved. Due to the proximity effect in the GN-diamond interface, intrinsic spin polarization occurs in the GN. 

Shiga and co-workers investigated the electronic properties of a GN nanoribbon with zigzag termination anchored to diamond surfaces [128]. They found that the unpaired spins at the edges of the GN nanoribbons tend to order ferromagnetically, which is a valuable property for the design of nanoelectronic devices operating with spin-polarized currents.

One of the most prominent applications of the GN is the FET. GN-based FETs with wavelength-dependent multiple optical inputs and one electrical output in response to the charge state of NV centers in diamond have been reported by Tzeng and co-workers [129]. The negatively charged NV center serves as the gate, with diamond being the gate dielectric, and produces an electric field to enhance the hole concentration in the GN channel. 

Improvement of the GN performance by replacing SiO_2_ with synthetic diamond was investigated by Yu et al. [130]. They investigated single-crystalline and UNCD substrates and concluded that the current carrying capacity of GN was improved in both cases. This study is technologically important as the UNCD is produced by a cost-friendly process compatible with conventional silicon technology.

Another aspect of real devices—the effect of vacancies on resistance at the GN-diamond interface was investigated [131]. The molecular dynamics simulations suggested that the interfacial resistance is driven by the distance of these point defects from the GN-diamond interface, implying that phonon dynamics around the vacancy reduce the interfacial barrier. Also, the static modification of the GN bonds at the interface at the vacancy position contributes to the final resistance.

A simple tunability of a bandgap is an extra challenge of the current all-carbon nanoelectronics technologies. A relation between the structure and electronic properties of quasi-2D hybrids composed of GN or hexagonal BN on hydrogen-terminated diamond heterostructures was reported by Mirabedini et al. [132]. The layers exhibit weak vdW interactions and tend to relax the induced strain via the formation of ripples. The GN on the H-terminated diamond (100) surface retains the semi-metallic characteristic. On the other hand, the BN on H-diamond is an indirect semiconductor with a bandgap of 3.55 eV due to Type-II band alignment. The authors concluded that the hexagonal BN maintains a defect-free interface on the hydrogen-terminated (100) diamond surface and provides tunability of electronic properties of surface-doped diamond-based FETs.

Muniz and co-workers presented a theoretical study of a bandgap opening in a unique configuration of sp^2^–sp^3^ carbon atoms at the inter-layer area of the hydrogenated twisted bilayer GN [133]. The special arrangement, with a structural pattern similar to the cubic or the hexagonal diamond, can be described as a 2D superlattice of ND-like crystals embedded in the GN layers. The predicted band gap’s value depends on the size of the diamond fragments incorporated in the superlattice unit cell.

Wan et al. [117] addressed fine-tuning of GN’s doping in GN-diamond Schottky heterojunctions. They investigated the electronic properties of GN transferred onto the boron-doped diamond with different surface termination; less usual n-doped GN was detected on the hydrogen and oxygen terminated surface. They concluded that GN doping is directly related to the surface dipole at the GN-diamond interface with a different surface termination. The value of the Schottky barrier height was found to be ~330 meV.

GN-diamond nanomaterials were also investigated for the applications, where the synergy of electronic, optical, and mechanical properties is required. Ueda et al. implemented GN-diamond heterojunctions as memristors and nonvolatile memory functions, in which photons switch the resistance-encoded states [134,135]. The hybrid junctions also exhibit wavelength selectivity for the resistance switching. The mechanism behind the high selectivity to the incident light can be attributed to the photoconductivity variation caused by redox processes on the GN-diamond interface.

## 5. Other Applications

Besides the nanoelectronics and tribology, the hybrid GN-diamond nanomaterials appeared to be very promising for other areas of applications, such as energy storage, ultra-fast and efficient detectors, and light sources and cutting or brazing technologies. In this section, the most important cases with broad application potential will be discussed.

### 5.1. Energy Storage—Supercapacitors

In general, the most efficient materials for supercapacitors are composed of nanowalls or nanotubes assembled in higher-order architecture. Banerjee et al. investigated boron-doped GN-diamond networks, where GN fragments are integrated into a 3D network of diamonds [136]. The all-carbon hybrid structure was produced by a one-step synthesis process. Interestingly, these hybrid GN-diamond nanomaterials show a high DLC value (0.43 mF·cm^−2^) and about 98% of electrode retention over 10^4^ cycles. The extraordinary performance was attributed to the unique morphology of the electrode material, in which the nanowalls are embedded in a highly conducting GN-like matrix. 

Another type of a 3D network composed of GN nanowalls on the CVD-grown diamond film was produced by a plasma-enhanced CVD on silicon substrates with ND particles, deposited as seeds [137]. During the electrochemical durability tests, the system revealed a low background current and a wide electrochemical potential window, which makes it a promising candidate for energy storage platforms.

Another type of sp^2^–sp^3^ hybrid supercapacitor material is the boron-doped GN/diamond composite [138]. It was synthesized by an electron-assisted hot-filament CVD technique and used for the fabrication of a model supercapacitor. The electrode revealed excellent performance up to high voltages, high energy density, and long-term stability.

Material’s porosity is also a significant factor influencing g performance of electrodes for energy storage applications. Flexible graphitic films with onion-like carbon nanoparticles sandwiched between thermally reduced GO or GN were prepared by Sun and colleagues [139], see Figure 4. The all-carbon films have a very high specific surface area (~420 m^2^·g^−1^) due to the presence of mesopores. The unique mesoporous morphology also permits their direct application without the need to add a polymer binder or a conductive additive. Composites of RGO-modified with ND particles were also used as efficient electrodes in electrochemical supercapacitors [140]. Graphite and silicon anodes coated with UNCD and GN nanowalls were tested as promising materials for anodes in lithium-ion batteries [90].

### 5.2. Detectors and Light Sources

The GN-diamond nanomaterials with convenient optical properties and appropriate photoresponse have been considered for applications as efficient photon and radiation detectors and sources across the broad range of the electromagnetic spectrum. 

A hybrid diamond-RGO platform for X-ray radiation detection was designed by Benfante and co-workers [141]. The system profits from the stability and fast response to the X-rays given by the polycrystalline diamond component and the low impedance of the RGO contacts. Also, the RGO on the diamond film can be easily patterned using standard lithographic procedures. All these characteristics are convenient for the technological implementation of this diamond-RGO system in large-scale X-ray beam monitors.

Another application related to ultra-fast detection of photons employs converting the optical signal of strong photon emitters via coupling to GN. For example, the NV centers in diamond undergo a non-radiative energy transfer to the GN, which can be detected with a picosecond time resolution via ultrafast transport measurements, exploitable in ultrafast electronics devices [142]. The GN also impacts the lifetime and intensity of the NV center fluorescence, as suggested by Liu et al. [29]. 

An essential application of GN-diamond application employs their excellent electron field emission (EFE) performance. Diamond-based cold cathode emission devices are typically composed of UNCD/GN nanocomposites or layered structures [121,143]. The GN is typically produced by high-temperature annealing of the UNCD films. 

The mechanism of the improvement of the plasma illumination properties and EFE in sp^2^–sp^3^ nanocarbon systems was investigated thoroughly by Shankaran et al. [144,145,146]. 

The nanomaterials with the best performance, such as ND/GN flakes, B-doped diamond/GN nanowalls, and few-layer GN/diamond nanorods, have a significant amount of the nanocarbon phases in direct contact. The authors concluded that GN-like phases in the grain boundaries of the ND crystallites form electron transport networks responsible for the superior EFE properties. Santos and co-workers also confirmed that the morpho-structural aspects are critical for the EFE behavior of GN/diamond hybrids [147]. Free-standing GN/diamond hybrid films [148] and n-type diamond/GN heterojunctions [149,150] were also recognized as very efficient nanomaterials for EFE applications.

A different type of photodetector based on a silicon/GN heterojunction with an sp^3^ carbon interlayer was designed by Yang and co-workers [151]. The authors concluded that incorporating the interlayer with a diamond-like arrangement of carbon atoms might be a universal strategy to construct hybrid carbon interfaces with high performance in next-generation optoelectronic devices.

The performance of the hybrid all-carbon nanostructures can be improved via synergy with other functional nanomaterials. For example, Huang et al. constructed UV photodetectors using a 2D array of ZnO nanotubes covered with UNCD and GN flakes [152], see Figure 5. The nanocarbons are responsible for the improved conductivity and formation of new energy levels in the conduction band of the ZnO, which help to enhance the transport of the charge carriers during UV illumination. These hybrid ZnO/nanocarbon photodetectors show unusually high photodetection sensitivity in the UV range of the spectrum.

Another type of hybrid nanocarbon-metal UV detector was demonstrated by Wei et al. [153]. The nanocomposite material shows high responsivity due to the heterojunction between the GN and microcrystalline diamond on a metal support. In a prototype photodiode, the GN (vertically aligned) serves as a transparent top electrode. Also, Yao et al. reported on a similar system for UV sensing—GN/diamond/thin metallic film fabricated on a flexible substrate [154].

Weinhold and co-workers fabricated a hybrid GN/diamond/dye device with an extremely thin architecture of all components [155]. The optically active layer contains terylene-based dye and melamine. The supramolecular assembly at the GN/diamond interface shows an absorption maximum at ~740 nm. Zhang et al. fabricated RGO/Au electrodes on polycrystalline diamond films grown on Si substrates [156]. These hybrid nanocarbon-metal nanostructures revealed excellent performance as alpha particles detectors. 

### 5.3. Materials’ Processing Technologies

The GN-diamond nanomaterials also play an important role in material processing. For example, GN, GO, and RGO attracted attention in cutting technologies. The addition of GN improves the performance of polycrystalline diamond compact (PDC) [157]. Therefore, a high-performance GN/PDC composite was prepared, and the hardness, wear-resistance, and electrical conductivity improved significantly, comparing to the pure PDC material. The strengthening mechanism in the GN/PDC composite is due to the well-known lubrication effect of the GN/diamond interface. In this vein, GN pallets are very efficient in suppressing the chemical wear of the tool in diamond cutting [158]. The GO helps improve the wear resistance of the epoxy resin-bonded diamond abrasive tools [159].

On the other hand, MLGN can help to reduce unfavorable thermal damage of diamonds in the current brazing practice [160,161]. It has also been reported that colloidal suspensions of GO mitigate carbon diffusion during diamond turning of steel [162].

Also, the rheology of mixed nanocarbon dispersions attracted some attention. Ilyas et al. addressed the use of GN/diamond materials in thermo-fluid technologies [163]. In this study, the hybrid suspensions contained an equal amount of diamonds with excellent thermal conductivity and GN nanoplatelets. The rheological behavior of these hybrids was explored with extensive machine learning methods. 

Interestingly, N-doped UNCD/MLGN hybrid films were employed in the construction of biocompatible carbon-based heaters for possible electrosurgical applications [164].

Another significant class of GN/diamond materials is epoxy composites. Jiang et al. Investigated the physical properties of such composites [165]. The materials exhibited high thermal and low electric conductivity. The presence of the GN component has an impact on several properties; it improves the dispersion of the diamonds in the polymer matrix, decreases the interfacial thermal resistance, and maintains excellent electrical insulation. 

The role of various carbon-based fillers in the epoxy thin-film composites was studied by Saw et al. [166]. The effect of loading and morphology of the diamond and GN fillers were explored. The nanocomposites coating diamond as a filler show excellent mechanical and thermal properties compared to the GN-containing composites. Both nanocomposites undergo a glassy transformation temperature with increasing filler content. 

Bittencourt et al. carried out a numerical simulation yielding the size distribution of GN clusters as a filler of a polymer aiming to obtain the percolation threshold [167]. The probability density of this distribution shows a universal complementary Fermi-Dirac behavior as a sign of a topological response. Using a tight-binding model for the transmission from the source to the drain, they obtain a smooth transition from an insulator to a conductor through a dirty metal state. As the dimensions of the array increase, the simulation shows a sharp non-metal-to-metal transition from a pure polymer into a pristine suspended GN layer.

Another type of nanocarbon-polymer composites containing poly(vinylidene fluoride) and functionalized-GN/ND fillers was produced by a simple solution method. The composite showed different thermal conductivities for the different ratios of the sp^2^ and sp^3^ phases in the hybrid filler [168].

Shul’zhenko and co-workers developed an HPHT process for the uniform dispersion of GN flakes in polycrystalline diamond, providing current-conducting diamond polycrystals [169].

The co-existence of GN and diamond-like phases and their mutual physiochemical interactions cannot be neglected in the growth procedures of the composite materials. For example, GO increases the chemical reaction rate of CVD diamond coating [170]. 

The diamond and GN-like nanocarbons also met on the way to scale up for industrial production of GO. A combination of fused-deposition-modeling-based 3D printing applied to a B-doped diamond material with a wide electrochemical potential window allowed to fabricate an efficient electrochemical reactor capable of producing electrochemically derived GO on a multiple-gram scale [171]. 

### 5.4. Catalysis

GN-diamond nanomaterials also serve as efficient catalysts or catalyst carriers. Lan et al. suggested using all-carbon GN/diamond nanomaterials for catalytic hydrochlorination of acetylene [172]. It is noteworthy that it is the first example of a metal-free catalyst with comparable performance to that of the 0.25% Au/C catalyst.

Another type of catalysts with an extreme surface area have been fabricated by a mild reduction/self-assembly hydrothermal method using GO dispersion as a precursor [173]. The obtained GN and RGO aerogels and GN or RGO aerogel/ND hybrids have been used as metal-free catalysts for oxidative dehydrogenation of propane. RGO aerogels without NDs show excellent productivity and selectivity due to a higher content of carbonyl-quinone groups due to the defective structure of the RGO. The GN aerogels with low ND content (2 wt%) about 18% higher activity than RGO aerogels ascribed to the sp^3^/sp^2^ ratio increase. The hybrid aerogels are freestanding, robust, and highly porous, suitable for applications in flow reactors.

The nanocarbon materials were also used in methanol fuel cells. ND/GN structures with a core-shell morphology were used as support for a platinum catalyst. The addition of GN is essential as the higher dispersion of the platinum catalyst was observed on ND/GN material featuring better catalytic activity and stability for methanol oxidation comparing to the platinum dispersed on the NDs [174].

### 5.5. Generation of Extreme Environments

GN and diamond were also used in several exotic applications. For example, water molecules were confined in nanobubbles at the hybrid interfaces of GN and diamond [175]. A molecular dynamics simulation revealed that the water molecules revealed much stronger hydrogen bonds for the high accommodation densities, giving rise to relaxation slow-down and blocking their rotation. In contrast, for lower densities of water molecules in the bubbles, the water dynamics is much enhanced. 

Lim and co-workers exploited the diamond in miniature high-pressure anvil nanocells [176]. They successfully generated pressure by heating the diamond crystal covered with a GN membrane; during the temperature increase, the membrane converted into GN nanobubbles. As in the previous case [155], those objects can also entrap water molecules. At high temperatures, the internal pressure in the GN-diamond voids dramatically increases because of the impermeability of GN. Interestingly, the water molecules accommodated in these nanoreactors etch the diamond surface giving rise to square-shaped holes. 

## 6. Summary and Outlook

GN-diamond and GN-ND nanostructures, heterojunctions, and derived hybrid nanosystems are a substantial all-carbon family of nanomaterials. The common denominators of the vast majority of technological and fundamental issues are the successful implementation of two-way communication between the graphene and diamond and the fragile interplay between the two border arrangements given by the two border cases of elemental carbon hybridizations (sp^2^ and sp^3^). Rational design and directed modification of the nanointerface and the implementation of functional nano links thus represent the major challenge in further developing GN-diamond-based nanomaterials and devices. Correspondingly, engineering the interface demonstrates an interesting topological problem—a reorganization can occur at partial atomic structures or in terms of topographic modulation. 

As can be seen even from the concise review, both sp^2^ and sp^3^ carbon allotropes offer many opportunities, especially when approaching the nanoscale dimensions, and many physical and chemical properties have already been utilized in practice. Note that the environment, which hides many pitfalls for implementing these nano-sized compounds into functional devices, should their exceptional properties not be compromised. Consequently, a next step—the bilateral communication of the nanocarbons and the surroundings, through highly efficient channels—the interfaces and the covalent links—must be promoted and rationalized in the near future.

## Figures and Tables

**Figure 1 nanomaterials-11-02469-f001:**
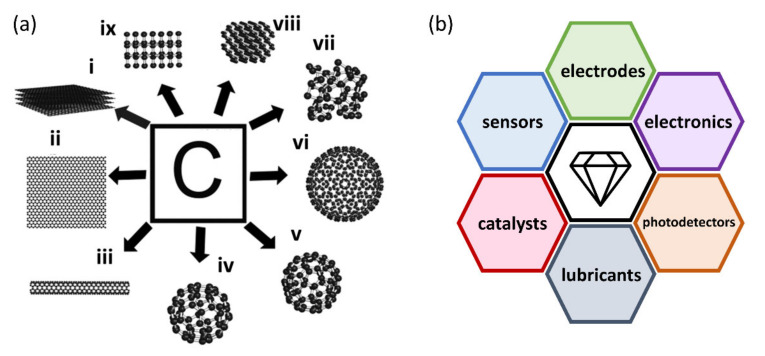
(**a**) Different carbon allotropes: (i) graphite, (ii) GN, (iii) carbon nanotube, (iv) fullerene C_60_, (v) C_70_, (vi) C_540_, (vii) amorphous carbon, (viii) lonsdaleite, and (ix) diamond. (**b**) Schematic presentation of the application areas of the GN—diamond nanomaterials.

**Figure 3 nanomaterials-11-02469-f003:**
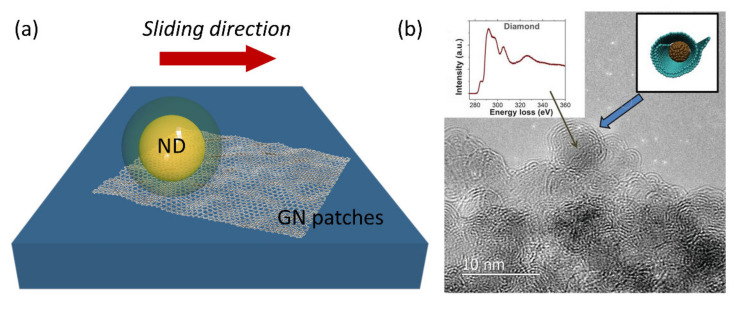
Superlubricity of NDs on GN due to formation of GN@ND scrolls. (**a**) schematic presentation of the superlubricity test; (**b**) TEM image of the resulting GN@ND scrolls with the electron energy loss spectrum recorded on the ND area. Reprinted with permission from [47]. Copyright 2015 The American Association for the Advancement of Science.

**Figure 4 nanomaterials-11-02469-f004:**
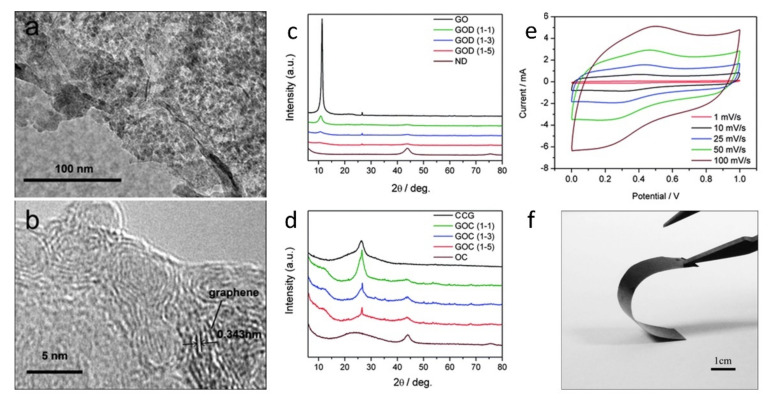
Example of GO—diamond composites for supercapacitors. Panels (**a**) and (**b**) present HRTEM images of the morphology at the nanoscale; panels (**c**) and (**d**) show typical XRD patterns of GO, ND particles, chemically converted GN films (CCG), onion-like carbon (OC), and mesoporous CCG/OC (GOC) composite films; panels (**e**) and (**f**) show cyclic voltammetry and photograph of the GOC film, respectively. Reprinted with permission from [139]. Copyright 2011 Royal Society of Chemistry.

**Figure 5 nanomaterials-11-02469-f005:**
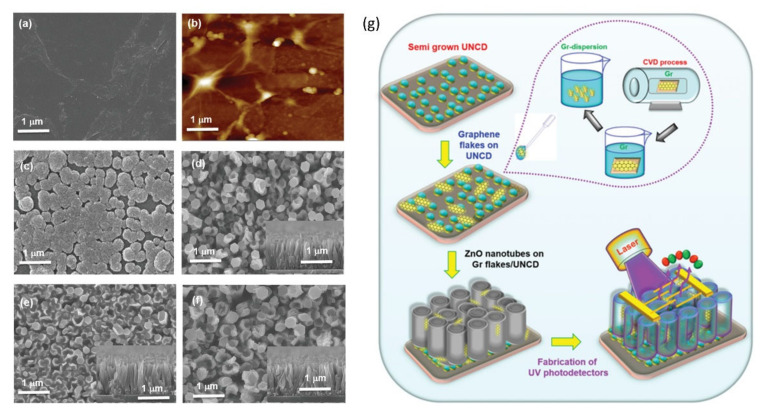
High-performance photodetector based on GN flakes, UNCD and ZnO nanotubes. (**a**) FESEM image of GN layer, (**b**) AFM image of GN layer, and (**c**) FESEM images of semigrown UNCD materials grown for 5 min. (**d**)–(**f**) FESEM images of UNCD-ZNT, GrF-ZnTs, and GrF-ZnTs/UNCD heterostructures. (**g**) Fabrication flow diagram of the UV photodetector. Reprinted with permission from [152]. Copyright 2020 John Willey and Sons.

**Table 2 nanomaterials-11-02469-t002:** Overview of graphene-diamond-based nanomaterials for various sensing applications.

System	Application	References
RGO/BDND	Protein sensor	[79]
Diamond/GN nanoplatelets/Pt nanoparticle hybrid	L-Glutamate electrochemical sensor	[80]
Ni nanoparticle-modified GN-diamond hybrid	Glucose sensor	[81]
Plasmonic gold nanostructure/diamond-like film	SERS sensor	[82]
Vertical GN sheets/separated papillary granules on ND film	ORR electrode	[83]
BD diamond/GN nanowalls electrode	Detection of explosives	[84]
GN/diamond electrode	Enantiomer recognition	[85]
GN/BDND electrode	Epinephrine detection	[86]
GO/BD-diamond electrode	Tetracycline detection	[87]
GN/BD-diamond electrode	Electrochemical sensing of trace Pb^2+^ in seawater	[88]
diamond, GN, and polyaniline/GC electrode	Electrochemical sensing of 2,4-dichlorophenol	[89]
N-doped UNCD/MLGN film	Electrochemical sensor (Ag^+^)	[90]
(Ag)GN-modified BD-diamond electrode	Electrochemical sensor of pesticides	[91,92]
GN/BD-diamond electrode	Electrochemical sensor of biomarkers	[93]
Few-layer GN/HPHT diamond electrode	Dopamine detection	[94]

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
