# Peer review of "Mixed sp2–sp3 Nanocarbon Materials: A Status Quo Review"

_nanomaterials, 2021, doi:10.3390/nano11102469_

Round 1

Reviewer 1 Report

Comments on nanomaterials-1338806

In this review, author summarized the status quo of graphene – diamond nanomaterials, including their composites, hetero junctions, and other hybrids for sensing, electronic, and other emergent applications. This review is meaningful which is helpful for the development of carbon materials. This review can be published on Nanomaterials after addressed the following questions:

  1. In this review, author listed almost all applications of current carbon materials. However, current content of this review is too broad. I suggest author make a targeted summary.
  2. The unambiguous definition (include classification, definition, and structure definition) of carbon materials should be agreed. Author need summarize current controversy on definition of carbon materials.
  3. In chapter 2, author summarized the research progress of graphene – diamond phase transformations at nanoscale. However, as another important performance modulation for carbon materials, the doping and modification also need be summarized. Moreover, some new developed carbon materials (such as carbon dots, graphene quantum dots, g-C3N4, C2N, C3N and C3N5) also need be summarized. Some relevant references: Nature Electronics, 2021, 4, 486; J. Am. Chem. Soc. 2019, 141, 5415; Advanced Materials, 2017, 29, 1605625; Nature Communications, 2015, 6, 6486; Adv. Mater. 2017, 29, 1702007; Adv. Mater. 2021, 33, 2005096; Small, 2020, 16, 2004621.
  4. The section number of “First principle calculations and modeling”, and “Experimental demonstration and growth mechanism” were wrong.
  5. The basic physical properties (such as band gap and carrier mobility) of carbon materials need be summarized.
  6. In chapter 4, author summarized the graphene – diamond interfaces and heterojunctions. However, the relevant physical scenario for the interfacial interaction in recent researches need be summarized.

Author Response

Reviewer 1

In this review, author summarized the status quo of graphene – diamond nanomaterials, including their composites, hetero junctions, and other hybrids for sensing, electronic, and other emergent applications. This review is meaningful which is helpful for the development of carbon materials. This review can be published on Nanomaterials after addressed the following questions.

  • Thank you very much for all your valuable suggestions and comments. Please, find the point-by-point reply below. All corrections are also given in the attached file.
  1. In this review, author listed almost all applications of current carbon materials. However, current content of this review is too broad. I suggest author make a targeted summary.
  • Thank you very much for this important comment, similar to one of the comments of the second reviewer. To explain, the main target of the review is mixed (hybrid) carbon nanomaterials, i.e., those containing sp2 (graphene and graphene-derived) and sp3 (diamond and diamond-like) carbon phases. Also, the exotic transitional phases (e.g., diamane) are considered. The original title was most likely misleading, suggesting a review of almost all carbon nanomaterials. Therefore, it was modified to: „Mixed sp2 – sp3 nanocarbon materials: a status quo review.“ The abstract was also modified to reflect better the content of the review.

  1. The unambiguous definition (include classification, definition, and structure definition) of carbon materials should be agreed. Author need summarize current controversy on definition of carbon materials.

  • Thank you very much for this comment. A paragraph on the definition of carbon materials supported by relevant references ([8] and [9]) was added to the Introduction.

  1. In chapter 2, author summarized the research progress of graphene – diamond phase transformations at nanoscale. However, as another important performance modulation for carbon materials, the doping and modification also need be summarized. Moreover, some new developed carbon materials (such as carbon dots, graphene quantum dots, g-C3N4, C2N, C3N and C3N5) also need be summarized. Some relevant references: Nature Electronics, 2021, 4, 486; J. Am. Chem. Soc. 2019, 141, 5415; Advanced Materials, 2017, 29, 1605625; Nature Communications, 2015, 6, 6486; Adv. Mater. 2017, 29, 1702007; Adv. Mater. 2021, 33, 2005096; Small, 2020, 16, 2004621.

  • Thank you very much for these comments. As explained above, the main target of the review was to address mixed (hybrid) carbon nanomaterials, i.e., those containing sp2 (graphene and graphene-derived) and sp3 (diamond and diamond-like) carbon phases. Nevertheless, paragraphs about the carbon nitride-based phases and carbon qunatum dots were added to the introduction based on the references kindly provided by the reviewer.

  1. The section number of “First principle calculations and modeling”, and “Experimental demonstration and growth mechanism” were wrong.

  • Thank you very much for pointing this out. Corrected.

  1. The basic physical properties (such as band gap and carrier mobility) of carbon materials need be summarized.

  • The values of the bandgap and typical carrier mobilities were added to the text together with some relevant references (graphene and other 2D nanocarbons: [6-7], [10-13]; nanodiamond: [25-28]).

  1. In chapter 4, author summarized the graphene – diamond interfaces and heterojunctions. However, the relevant physical scenario for the interfacial interaction in recent researches need be summarized.
  • Thank you very much for this valuable comment. A paragraph on the physical scenario of the graphene-diamond interfaces and heterojunctions was added to section (chapter) 4.

Reviewer 2 Report

This manuscript reviews a current status of graphene-related diamond and diamane. In particular, the nanodiamond transformed from graphene is an interesting issue, so this topic is timely. I have just several corrections and suggestions that the author should address.

“graphene – diamond nanomaterials” term in the title and text is quite ambiguous. The author should define this term and find a more appropriate term.

Authors need to correct Table 1, Ref [22] provided HRTEM and EELS results to prove F-diamane. This is crucial evidence of the diamane formation converted from bilayer graphene.

Minor corrections:

On line 191, SP2 and SP3: numbers should be superscripted

In Table 1, HR TEM should be HRTEM without a blank in the abbreviation.

Author Response

Reviewer 2

This manuscript reviews a current status of graphene-related diamond and diamane. In particular, the nanodiamond transformed from graphene is an interesting issue, so this topic is timely. I have just several corrections and suggestions that the author should address.

  • Thank you very much for the positive evaluation and valuable comments, which helped to improve the manuscript significantly. Please, find the point-by-point response below. The corrections are also visible in the attached file.

“graphene – diamond nanomaterials” term in the title and text is quite ambiguous. The author should define this term and find a more appropriate term.

  • Thank you very much for this valuable comment. I have to admit that finding an appropriate title for the review was a challenge. The main idea was to address all important sp2 – sp3 based carbon nanomaterials, where the sp2 is graphene or graphene-derived material and sp3 is the diamond. Consequently, the transitional sp2 – sp3 phases occurring at the interfaces were also included. I suggest changing the title to „Mixed sp2 – sp3 nanocarbon materials: a status quo review“

Authors need to correct Table 1, Ref [22] provided HRTEM and EELS results to prove F-diamane. This is crucial evidence of the diamane formation converted from bilayer graphene.

  • Thank you very much for pointing out this inconsistency. Table 1 was corrected.

Minor corrections:

On line 191, SP2 and SP3: numbers should be superscripted

  • Thank you very much for pointing this out. Corrected.

In Table 1, HR TEM should be HRTEM without a blank in the abbreviation.

  • Thank you very much for pointing this out. The abbreviation „HRTEM“ was updated in the whole text.
